# Statistical potentials from the Gaussian scaling behaviour of chain fragments buried within protein globules

Stefano Zamuner[1], Flavio Seno[2,3], Antonio Trovato[2,3]*

**1** Institute of Physics, École Polytechnique Fédérale de Lausanne, Lausanne, Switzerland, **2** Department of Physics and Astronomy "G. Galilei", University of Padova, Padova, Italy, **3** INFN, Sezione di Padova, Padova, Italy

* antonio.trovato@unipd.it

## Abstract

Knowledge-based approaches use the statistics collected from protein data-bank structures to estimate effective interaction potentials between amino acid pairs. Empirical relations are typically employed that are based on the crucial choice of a reference state associated to the null interaction case. Despite their significant effectiveness, the physical interpretation of knowledge-based potentials has been repeatedly questioned, with no consensus on the choice of the reference state. Here we use the fact that the Flory theorem, originally derived for chains in a dense polymer melt, holds also for chain fragments within the core of globular proteins, if the average over buried fragments collected from different non-redundant native structures is considered. After verifying that the ensuing Gaussian statistics, a hallmark of effectively non-interacting polymer chains, holds for a wide range of fragment lengths, although with significant deviations at short spatial scales, we use it to define a 'bona fide' reference state. Notably, despite the latter does depend on fragment length, deviations from it do not. This allows to estimate an effective interaction potential which is not biased by the presence of correlations due to the connectivity of the protein chain. We show how different sequence-independent effective statistical potentials can be derived using this approach by coarse-graining the protein representation at varying levels. The possibility of defining sequence-dependent potentials is explored.

## Introduction

Proteins are linear flexible hetero-polymers, made up of 20 different natural amino-acid species [1]. Most natural proteins in solution have roughly compact shapes, and thus are usually referred to as globular proteins. The fundamental fact about globular protein sequences is their ability to attain a compact native three-dimensional folded conformation in physiological conditions [2].

The biological functionality of proteins is intimately related to their native structures and to the dynamical properties encoded in them [3]. Quantitative theoretical modeling requires in principle a detailed description at atomic level, for example to take accurately into account the subtle yet dramatic effects that can be brought about by a single residue mutation.

**Competing interests:** The authors have declared that no competing interests exist.

On the other hand, processes such as protein self-assembly and aggregation, involve time scales and system sizes which are currently unattainable by atomistic models [4]. Several schemes were thus developed to coarse-grain the representation of protein structures, and of the physical interactions between the representing entities, at a low resolution level [5].

The surprising success of coarse-graining approaches in computational protein science is related to the presence of robust qualitative emergent properties in protein systems, amenable to prediction by low resolution models [6]. For example, the native topology both shapes equilibrium fluctuations and determines folding and unfolding pathways, allowing for successful predictions by structure-based coarse-grained models [7].

An even more remarkable example of successful coarse-graining is the use of statistical potentials, as both effective interaction potentials to be used in folding simulations [8–10], and scoring functions employed in different contexts such as protein structure and function prediction [11], "de novo" protein design [12], model quality assessment [13, 14], aggregation propensity prediction [15–17], protein-protein interactions [18–22], prediction of binding affinities and of stability changes upon mutations [23–26], and many others. Statistical "knowledge-based" potentials can be introduced at different coarse-graining levels, including atomic resolution (in this case, the coarse-graining is due to solvent molecules being integrated out). They renounce a physics-based description of the effective interactions between representing entities; interactions are instead parametrized using the statistics empirically collected from the Protein Data Bank (PDB) [27].

In paradigmatic examples [28, 29], "contact statistical potentials" evaluate the effective interaction between a pair of amino acid residues based on the observed frequency of contacts between that pair in PDB structures. This approach can be generalized to many different observables, such as solvent accessibility, backbone dihedrals, orientation-dependent or many body interactions [30–34]. The conversion of empirical frequencies into an energy function is normally done employing Boltzmann inversion, as originally suggested by Sippl for pairwise "distance dependent potentials", in analogy to the pairwise potentials of mean force [35]. Complementary potentials are typically estimated separately, to be then combined together, for interactions either short-range or long-range along the chain [36], with the aim of correctly capturing local structure elements. In state-of-the-art approaches several different statistical potential terms, each related to a different observable, can be combined together, optimizing the relative weights by means of supervised learning techniques [14, 37–40].

A crucial element in the definition of statistical potentials via Boltzmann inversion is the choice of a "reference state". The probability distribution observed in the latter is used to normalize the statistics collected over the PDB structures for a given residue pair. The reference state should then be taken as an ensemble of protein-like structures where no specific direct interactions between amino acids are present. A simple choice is to consider the ensemble of all residue pairs from the PDB structures [41], but still many different recipes are possible to define the reference state [42–45]. Beside the uncertainty in the reference state definition, the very use of Boltzmann inversion for the statistics collected from different PDB structures was extensively debated [46, 47], in particular with reference to chain connectivity. The Boltzmann inversion has been justified by using information theory arguments within a Bayesian approach [48]. In this context, statistical potentials are considered as statistical preferences that can be obtained "a posteriori" from empirical data, whereas the reference state contains the "a priori" information about the system.

In this work, we propose to use a reference state for deriving pairwise distance dependent potentials based on purely polymer physics considerations. Our strategy can be used at different levels of coarse-graining.

In particular, we use the fact that a data set of protein "fragments", when collected and properly filtered from PDB structures, exhibit Gaussian statistics, the one expected for ideal

chains in the absence of any interaction. This property had been already uncovered by Banavar et al. [49], who found that fragments buried within globular proteins obeys on average the same Flory theorem [50, 51] derived for polymer melts, that is concentrated solutions of different chains. The same theorem has been shown to hold for fragments buried in the interior of single compact polymer chains, when selected with appropriate constraints [52]. The Flory theorem states that, for polymer chains from within a dense polymer melt, excluded volume repulsion is effectively canceled by solvent-mediated attractive forces between the monomers. Therefore the chains exhibit statistics which are characteristic of random walk behavior.

The first purpose of this work is to confirm the existence of a Flory regime for buried protein fragments when a much larger data set of proteins is considered. We then take advantage of this fact by using the ensuing Gaussian reference state in order to obtain an unbiased estimation of a distance dependent effective interaction potential between aminoacids [41, 48, 53]. The statistical potentials estimated with this strategy could be either sequence independent or sequence dependent.

In the first part of the paper, by analyzing a data set of 7793 non-redundant globular proteins, we confirm that the Flory theorem holds for compact native structures with a good degree of accuracy. This is achieved by showing that the properly rescaled distributions of the fragment end-to-end distances collapse to the same Maxwell distribution when fragment lengths larger than $m_{min}$ = 70 and smaller than $N^{2/3}$, where $N$ is the length of the protein chain, are considered. The upper cut-off is introduced in order to select buried fragments [52]. The lower one is instead necessary to achieve a uniform Kuhn length. Our results extend the findings of Ref. [49], showing that the Gaussian scaling holds for fragments in a larger range of sizes, provided a non uniform Kuhn length is considered.

As a consequence of the validity of the Flory theorem, we can assume that within protein globules the excluded volume repulsion is on average effectively canceled by solvent-mediated attractive forces between the monomers. We therefore interpret systematic deviations from the expected Gaussian behavior, which are visible in the short spatial range regime as an effective intra-molecular interaction, that can be then considered unbiased by the spurious correlations due in general to the chain constraint, to local conformational preferences, or to interior-exterior partitioning effects.

Therefore, we devoted the second part of this manuscript to exploit the feasibility of this idea, by estimating a sequence-independent effective potential based on the statistics observed for buried protein fragments at different coarse-graining levels: CA-based, all heavy atoms, all atoms (including hydrogen atoms). The estimated potentials consistently change in the three cases. In particular, a power law repulsive term is present at short length whereas the potential vanishes beyond ≈20 Å in all cases. Well defined minima with negative energy are present for the atomistic resolutions for distances compatible with the sum of Van der Waals radii or with hydrogen bond geometry.

If the analysis is repeated by classifying the protein segments according to the amino-acid types which occupy the first and the last position along the fragment, we can have a direct measure of the effective interactions between amino-acid types, as a function of the distance. This method could be of great interest for a wide range of applications in protein physics.

## Materials and methods

### Dataset

Our database of reference is Top8000 [54], which contains a set of 7957 high-resolution protein structures. The dataset has been filtered by excluding those structures of length $N$ that do not exhibit a globular shape, i.e. whose gyration radius $R_g(N)$ does not scale as $N^{\frac{1}{3}}$. In order to

achieve this we fit experimental data with the relation $R_g(N) = aN^{\frac{1}{3}}$ and discard all the structures that fall more than three standard deviations apart from the fitted curve. 164 structures have been discarded in this phase.

The tangent vector to each residue has been computed as the difference between the coordinates of the subsequent and the previous residues along the chain. We reported the average cosine of the angle between the tangent vectors of pairs of residues as a function of their separation $m$ along the chain. As the tangent-tangent correlation goes to zero when $m \sim 30$, we decided to exclude from our analysis chain fragments shorter than 30 amino acids.

We therefore split all the protein chains in fragments of length $30 \leq m \leq N^{\frac{2}{3}}$ and grouped them by length. All other fragments have been discarded.

## Reference and empirical distributions

We measured the end-to-end distance $R$ of all fragments of given length $m$.

We fitted the rescaled data at fixed $m$ with a Maxwell distribution

$$\mathcal{M}_m(R, b) = \frac{4\pi R^2}{\left(\frac{2}{3}\pi b^2\, m\right)^{\frac{3}{2}}} \exp\left(-\frac{3R^2}{2b^2\, m}\right),$$ (1)

with a single free parameter $b$ (the scale, a.k.a. Kuhn's length) by using the scipy.stats python package and a maximum likelihood fit.

The empirical distribution $\mathcal{E}_m(R, w)$ has been obtained by employing a Kernel Density Estimation (KDE) with a Gaussian kernel:

$$\mathcal{E}_m(R, w) = \frac{1}{M} \sum_\sigma \frac{1}{\sqrt{2\pi}w} \exp\left(\frac{(R - r_\sigma)^2}{2\, w^2}\right).$$ (2)

The sum is extended over all $M$ values $r_\sigma$ in the dataset of end-to-end distances of fragments of length $m$. We used cross validation in order to establish the optimal kernel bandwidth $w$ for fragment lengths $m \in \{42, 48, 60, 64, 66, 72, 78, 84, 92\}$. We divided every set of end-to-end distances of fixed fragment length in five groups: an empirical distribution was computed using the data of four groups. The width of the Gaussian kernel was therefore adjusted in order to maximize the likelihood that the data from the fifth group was obtained from the same empirical distribution.

In order to estimate the optimal bandwidth for all other datasets, we assumed the relation $w = an^s$ between the bandwidth $w$ and the number of points $n$ in the dataset. We fitted the parameters $a$ and $s$ by minimizing the RMSD with the cross-validated bandwidths (see S1 Fig in S1 File).

## Potential

For every sequence separation $m$, we estimated the potential as a potential of mean force depending on the distance $R$ between two residues, using the ensemble of buried fragments selected as described in the Dataset subsection. Following a seminal approach [35], we assume $R$ to be distributed according to the Boltzmann distribution

$$P_m(R) = \frac{1}{Z_m} \exp\left(-\frac{F_m(R)}{\kappa_B T}\right),$$ (3)

where $\kappa_B$ is the Boltzmann constant, $T$ is the temperature at which thermodynamic equilibrium is assumed to hold, and $Z_m$ is the canonical partition function. In what follows, we

assume $\kappa_B T = 1$ for simplicity. One should keep in mind that $F_m(R)$ is in fact an effective free energy, since it is obtained by coarse-graining other degrees of freedom, including for example the ones associated to solvent molecules. Boltzmann inversion than implies

$$F_m(R) = -\ln P_m(R) - \ln Z_m \ . \tag{4}$$

The potential of mean force is than defined as a free energy difference $\Delta F_m(R)$ with respect to the ideal reference state, characterized by the Maxwell distribution $\mathcal{M}_m(R, b)$ and the partition function $Z_m^*$, with the scale $b$ determined as described in the previous subsection:

$$\Delta F_m(R) = -\ln \frac{P_m(R)}{\mathcal{M}_m(R, b)} - \ln \frac{Z_m}{Z_m^*} \ . \tag{5}$$

The term with the (unknown) partition function ratio can be neglected since it does not depend on $R$, and as a proxy of the Boltzmann distribution $P_m(R)$ we use the empirical distribution $\mathcal{E}_m(R, w)$, evaluated using KDE, with bandwidth $w$ optimized as described in the previous subsection. This leads finally to our estimation for the statistical potential:

$$V_m(R|b, w) = -\ln \left( \frac{\mathcal{E}_m(R, w)}{\mathcal{M}_m(R, b)} \right) \ . \tag{6}$$

Note that Eq (6) defines an average pairwise residue-residue sequence-independent potential. Both the empirical and the Maxwell distributions entering Eq (6) are obtained based on the statistics of all fragments in our filtered dataset. The pairwise decomposition of the total score for a whole protein,

$$V_{tot} = \sum_{i<j} V_{|j-i|}\left(R_{ij}|b, w\right) = -\ln \left( \frac{\prod_{ij} \mathcal{E}_{|j-i|}(R_{ij}, w)}{\prod_{ij} \mathcal{M}_{|j-i|}(R_{ij}, b)} \right) , \tag{7}$$

is in general an approximation since it neglects correlations between different residue pairs. The major point in our analysis is related to the absence of effective interactions that is actually realized in the reference state. This implies that the pairwise decomposition is exact for the reference state (the denominator in the r.h.s of Eq (7)).

As we show that the potential does not depend on $m$, we finally computed $V^*(R)$ as the average over all possible fragment lengths of $V_m(R|b, w)$ in the Flory regime, $70 \leq m \leq 90$. Note that both parameters $b$ and $w$ depend on fragment length $m$.

We fitted the short range repulsive part of the potential by minimizing the root mean square deviation between the logarithm of $V^*(R)$ and a linear function.

## Sequence-dependent analysis

We repeated the previous analysis while filtering the fragments depending on the amino acids types at their ends, so that both the empirical distribution $\mathcal{E}_m(R, w)$ and the Maxwell distribution $\mathcal{M}_m(R, b)$ are obtained from such restricted fragment sets. To increase available statistics, the average $\overline{V}(R)$ of $V_m(R|b, w)$ is now taken over all fragment lengths $30 \leq m \leq 90$ with a Gaussian reference state.

## Results

In order to assess the hypothesis that the Flory theorem holds for fragments buried in the interior of globular proteins, we analyzed a large data-set of 7793 globular proteins. This protein ensemble was obtained by refining the TOP8000 data-bank [54] after removal of the non globular structures as explained in Methods. In data-set pruning, each protein is represented as a

polymer whose monomers are placed in the $C_\alpha$ atomic position of the $N$ amino-acids. The logarithmic plot of the radius of gyration of these polymers versus their length $N$ is shown in S2 Fig in S1 File for the full TOP8000 data-bank. The proteins in the final pruned dataset (highlighted in S2 Fig in S1 File) have been selected in such a way that their radius of gyration scales as $N^{1/3}$, as expected for globular proteins.

## Long enough buried protein fragments follow Gaussian statistics: The thermal exponent

To investigate the validity of the Flory theorem, we analyzed an ensemble of protein fragments of different lengths, extracted from the pruned database. For any given chain of length $N$ we considered only fragments with length $m < N^{2/3}$ so that they belong to a part of the protein which is likely to be far from globule boundaries and thus buried within the globule interior [52]. Fig 1a shows in a logarithmic scale the behavior of the average end-to-end distance $R$ of such fragments as a function of their length $m$, when the end-to-end distance is evaluated using $C_\alpha$ backbone atoms.

The Flory regime requires a scaling law $R \sim m^\nu$ with $\nu = \frac{1}{2}$. Our data for CA end-to-end distances show that such a regime is valid only for the longer fragments, approximately when $m > 70$. This can be explained by the presence of secondary structures that introduce a strong bias in the scaling behavior for short fragments. This behavior can be understood by looking at S3 Fig in S1 File which shows the average tangent-tangent correlation as a function of sequence

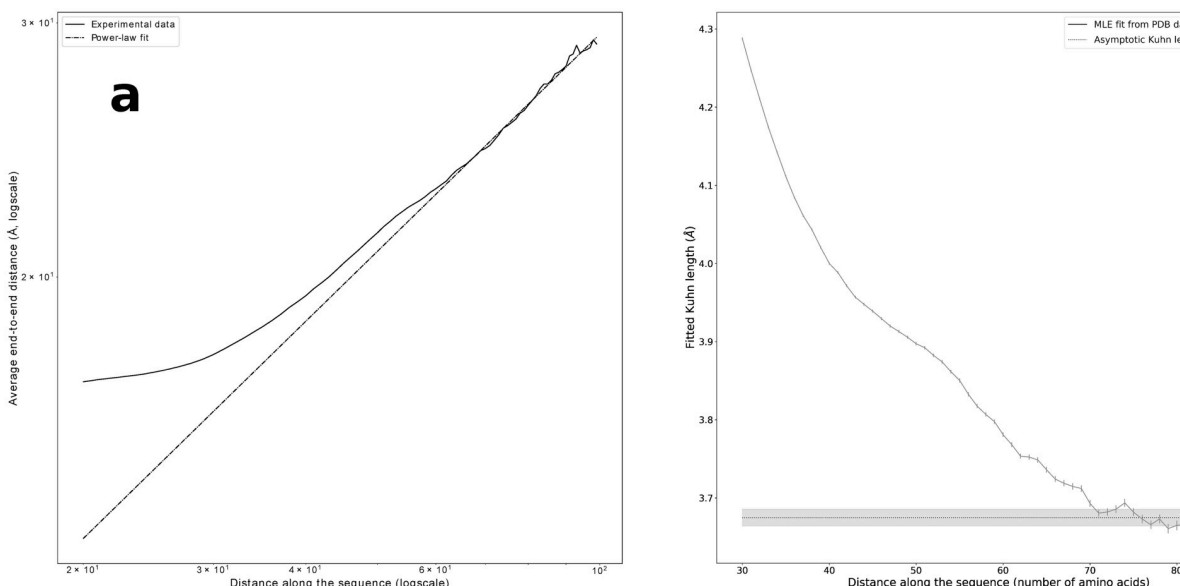

**Fig 1. Gaussian statistics for buried fragments: The thermal exponent and the Kuhn length.** (**a**) Log-log plot of the average CA end-to-end distance $R$ of protein fragments versus fragment length $m$. The plot was obtained by averaging over all fragments of length $m$ from the data set selected as shown in S2 Fig in S1 File. For any given $m$, $R$ was determined as the average over all fragments of that length in proteins whose overall lengths are larger than $m^{\frac{3}{2}}$, in order to consider only fragments likely to be buried in the globule interior [52]. The error bars are of the order of the size of the symbols. The Flory regime, e.g. $R \sim m^{\frac{1}{2}}$ is reached when $m \geq 70$. For $m > 90$ the statistical analysis deteriorates due to the fast decrease of available data with increasing $m$. (**b**) The Kuhn length $b$, obtained by maximizing the likelihood to Maxwellian distributions (1) of the empirical CA end-to-end distance data, plotted versus the length $m$ of the protein fragments considered in the statistical analysis. The error bars were estimated based on the Fisher information evaluated at $b(m)$ (see main text). The values of $b$ decrease monotonically and reach a plateau in the region $70 \leq m \leq 90$. The plateau uniform value is estimated to be $b^* = 3.67 \pm 0.01$ Å. Only in this region all the rescaled empirical distributions collapse (see Fig 3), thereby showing the existence of the Flory regime. The number of fragments in the ensembles which are analyzed decreases with $m$ as well. For $m > 90$ the ensemble population becomes too small to allow good estimations. Figure drawn with python package matplotlib, version 3.4.1. URL https://pypi.org/project/matplotlib/.

separation along the chain. For short sequence separations the direction of the chain is highly correlated reflecting the existence of short, effectively one-dimensional, rigid motifs such as $\alpha$-helices and $\beta$-strands. On the other hand, the sharp anti-correlation minimum at $m = 13$ reveals a bending propensity in the opposite direction. This finds its counterpart in the almost flat behaviour of the average end-to-end distance in Fig 1a for $m \gtrsim 30$. This picture is confirmed by noticing that the value at which the correlation function reaches again zero ($m \sim 25$) is almost twice as much as the value at the anti-correlation minimum, suggesting that for $m \sim 25$ protein chains are expected to loop back on themselves significantly more than for other values of sequence separation. In fact, the above observation is consistent with the peak described in [55] for the probability of loop formation. This analysis suggests that the Flory regime can not be observed for short fragments because of the effects induced by secondary structures.

## Long enough buried protein fragments follow Gaussian statistics: Maxwellian distributions for end-to-end distances

In order to investigate in more detail the existence of a Flory regime we studied whether the end-to-end distance $R$ for fragments of length $m$ follows the Maxwell distribution described by

$$\mathcal{M}_m(R, b) = \frac{4\pi R^2}{\left(\frac{2}{3}\pi b^2 m\right)^{\frac{3}{2}}} \exp\left(-\frac{3R^2}{2b^2 m}\right), \tag{8}$$

where the scale parameter $b$ refers to the distance between consecutive monomers in an ideal Gaussian chain, namely the Kuhn length of the polymer. The distances between residues are computed in three different ways: as the distance between their $\alpha$-carbons, as the minimum distance between any atom of the two residues and as the minimum distance between any heavy atom of the residues. We will refer to these three different levels of coarse-graining as CA ($\alpha$-carbon level), HH (hydrogen atoms level) and HV (heavy atoms level) respectively. For all three coarse-graining levels of description, we fitted the Kuhn length $b$ of a Maxwell distribution to maximize the likelihood that the empirical data have been drawn from it. This is done separately for any given $m$, so that the optimized Kuhn length $b(m)$ depends on fragment length $m$. In Fig 1b we plot the optimized $b$ as a function of $m$ for CA end-to-end distances. The standard deviations $\sigma_{b(m)}$ associated to the maximum likelihood estimators $b(m)$ are shown in Fig 1b as error bars. Based on the Fisher information evaluated at $b(m)$, they were estimated as $\sigma_{b(m)} = b(m)/\sqrt{6n(m)}$, where $n(m)$ is the number of fragments in the dataset for a given $m$ (see Table 1). The values of $b$ decrease towards a plateau beginning approximately at $m = 70$. For $m > 90$ the values of $b$ change dramatically as a consequence of the poorer statistics (see Table 1). At the plateau we estimate the $m$-independent Kuhn length $b(m) \simeq b^* = \sum_{m=70}^{90} b(m)/21 = 3.67 \pm 0.01$ Å for CA end-to-end distances. The standard deviation of $b^*$ is estimated accordingly. Similar results are obtained for HV and HH as well, as shown in S4 Fig in S1 File. The plateau values estimated for the Kuhn length are $b^* = 3.35 \pm 0.01$ Å for HV and $b^* = 3.27 \pm 0.01$ Å for HH.

Fig 1b shows that the estimated Kuhn length is higher than in the plateau for lower values of $m$. This could explain the discrepancy with the higher value ($b^* = 3.75$ Å) obtained in [49], as a different range of values of $m$ was used in that work.

Empirical probability distributions are inferred by raw data using Kernel Density Estimation (KDE), with kernel bandwidth estimated separately for each $m$ by a maximum likelihood approach through a cross-validation procedure. The whole methodology is explained in detail in Methods.

**Table 1. Length dependent statistics of buried fragments.**

| fragment length | number of fragments in the dataset |
| --- | --- |
| 20 | 1640293 |
| 30 | 1352751 |
| 40 | 991104 |
| 50 | 563178 |
| 60 | 276421 |
| 70 | 120448 |
| 90 | 24967 |

Number of buried ($m < N^{2/3}$) fragments in the pruned (see S2 Fig in S1 File) dataset as a function of fragment length.

In Fig 2, the empirical CA end-to-end distance probability distributions for four different fragment lengths ($m$ = 30, 50, 70, 90) are shown together with their best Maxwellian fits. Similar plots for HV and HH end-to-end distance distributions are shown in S5 and S6 Figs in S1 File. The competing effects of increasing $m$ and of $b(m)$ decreasing with $m$ are both visible in Fig 2.

It is interesting to observe that, as shown in Fig 2, S5 and S6 Figs in S1 File, the Gaussian behavior of the end-to-end distances is mostly preserved even in a broader range of fragment lengths, $30 \leq m \leq 90$, but since $b(m)$ is not uniform for $m < 70$, we cannot talk about a Flory regime in that range. Nevertheless, the existence of a Gaussian reference state can be fruitfully exploited to derive an effective statistical potential in the full $30 \leq m \leq 90$ range.

In the region with uniform Kuhn length $b$ ($70 \leq m \leq 90$) empirical distributions can be collapsed by rescaling the end-to-end distances by $\sqrt{m}$ and multiplying the probability distributions by the same quantity, as shown in Fig 3 for CA. This graph vividly shows the existence for globular proteins of a range of fragment lengths, in which their statistics is well described by Gaussian ideal chains with a uniform estimated Kuhn length, $b^* = 3.67$ Å, close to the average distance, $\simeq 3.8$ Å, found between consecutive $C_\alpha$ atoms in protein native structures. The value of the Kuhn length may appear surprisingly low. We can rationalize this fact by looking at the average tangent-tangent correlation function (S3 Fig in S1 File). Since the value of the Kuhn length is related to the integral of the latter over all sequence separations [56], the presence of a sharp minimum at a negative correlation, implying a significant negative contribution to the integral, is at least consistent with the low value that we find for the Kuhn length. A data collapse of similar quality can be obtained for both HV and HH, as shown in S7 Fig in S1 File. S7 Fig in S1 File also shows that the collapsed empirical distributions are more skewed with respect to the reference Maxwell distribution in the HV and HH cases.

## Statistical potentials with a Gaussian reference state: Sequence independent effective interaction

The Maxwell distribution (8) fits very well CA experimental data for large values of end-to-end distance, when the full cancellation of competing interactions, e.g. attractive and excluded volume, is effectively occurring. For short distances however, as we can see from Fig 3, there are important deviations from the ideal distribution. These are expected, since for an ideal chain excluded volume is absent even at short range, whereas for real protein chains it is anyway present. We then propose to use deviations from the ideal Gaussian behavior as a proxy of the effective short range interactions between protein residues.

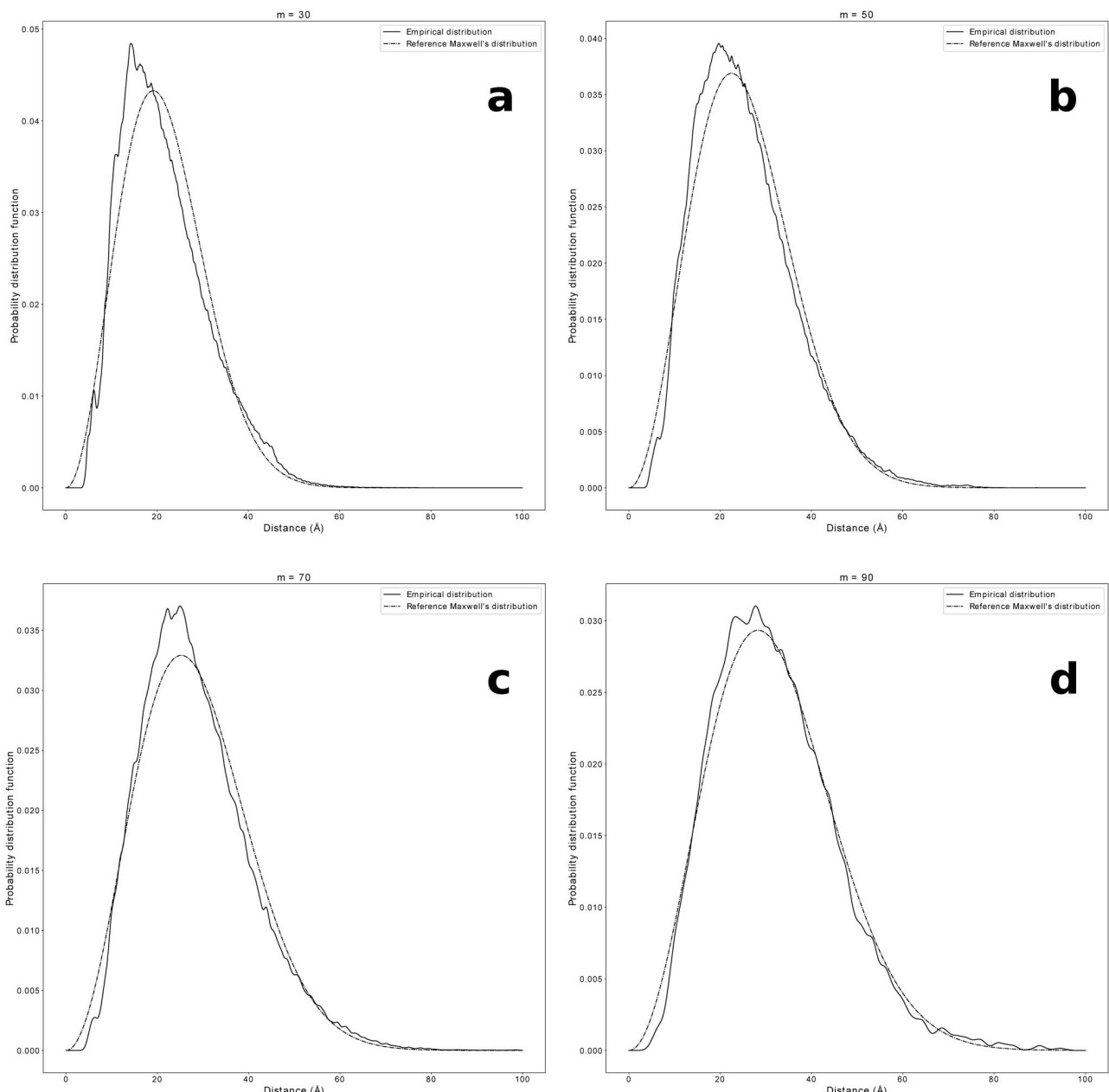

**Fig 2. Gaussian statistics for buried fragments: End-to-end distance.** The CA end-to-end distance probability distributions for four different fragment lengths, (**a**) $m = 30$, (**b**) $m = 50$, (**c**) $m = 70$, (**d**) $m = 90$, are shown together with their best fits to the Maxwell distribution (8). The parameters $b$ used in the plots are obtained maximizing the likelihood that the empirical data belong to the Maxwell distribution (8). The value of $b$ decreases with $m$ and reaches a plateau for $70 \leq m \leq 90$, corresponding to the Flory regime (see Fig 1b). Figure drawn with python package matplotlib, version 3.4.1. URL https://pypi.org/project/matplotlib/.

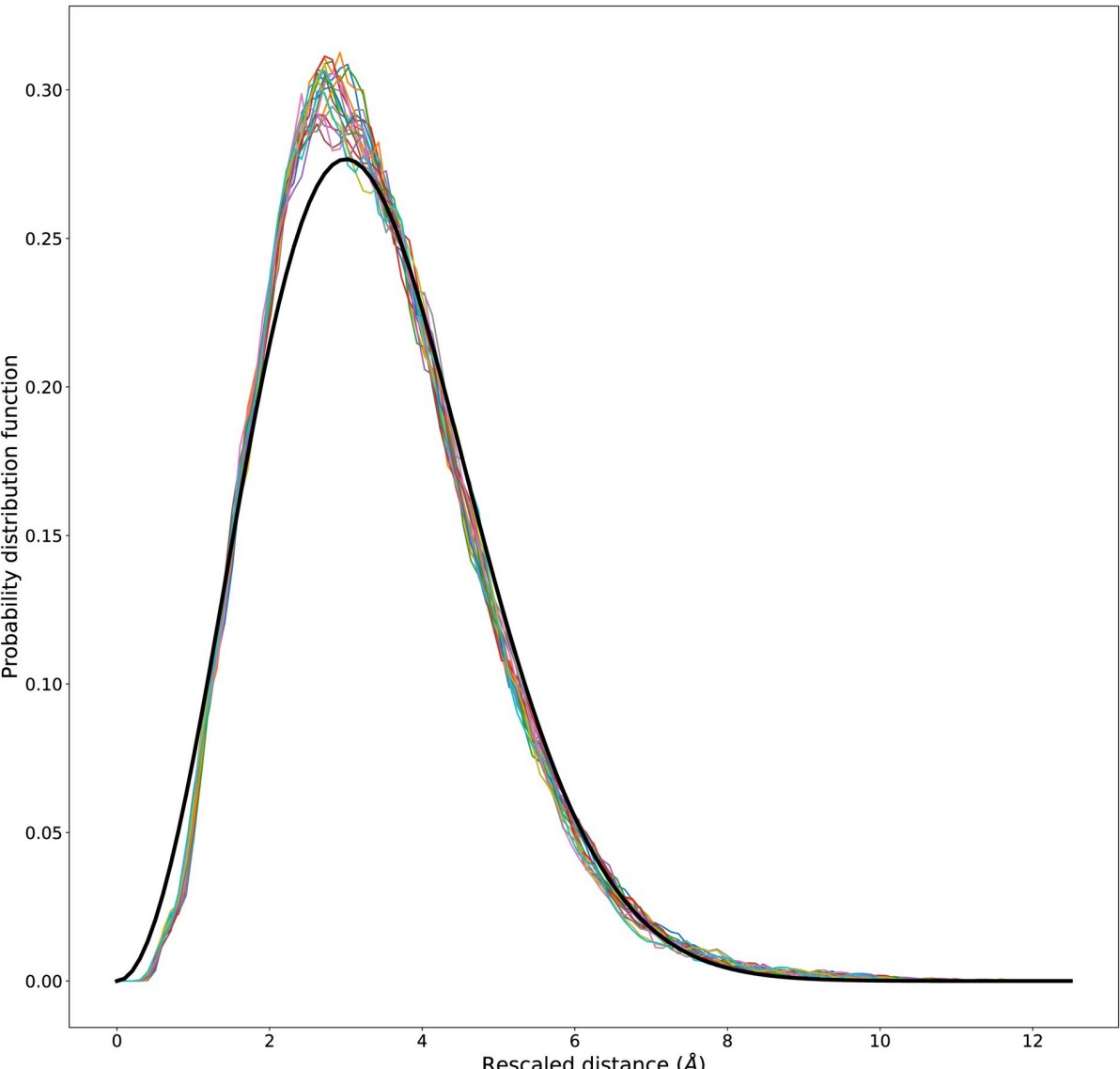

**Fig 3. Data collapse of end-to-end distance distributions in the Flory regime for buried fragments.** The rescaled empirical probability distributions as a function of the rescaled length $R/m^{1/2}$ for $70 \leq m \leq 90$. All curves collapse rather well together. The reference Maxwell distribution (8) evaluated for the plateau scale parameter $b^* = 3.67$ Å is shown for comparison. A significant deviation appears only for small distances and is due to the effect of excluded volume that at very short range can not disappear for real protein chains. It is worth to notice that, despite the presence of secondary structures, the value of $b$ is close to the average distance, $\simeq 3.8$ Å, found between consecutive $C_\alpha$ atoms in protein native structures. Figure drawn with python package matplotlib, version 3.4.1. URL https://pypi.org/project/matplotlib/.

To test this hypothesis, we define through Boltzmann inversion a sequence independent statistical potential for any given fragment length $m$, as minus the logarithm of the ratio between the empirical probability density (already shown in Figs 2 and 3 for different fragment lengths) and the reference Maxwell distribution:

$$V_m(R|b, w) = -\ln\left(\frac{\mathcal{E}_m(R, w)}{\mathcal{M}_m(R, b)}\right) ; \tag{9}$$

where $\mathcal{E}_m(R, w)$ is the empirical end-to-end distance distribution for fragments of length $m$,

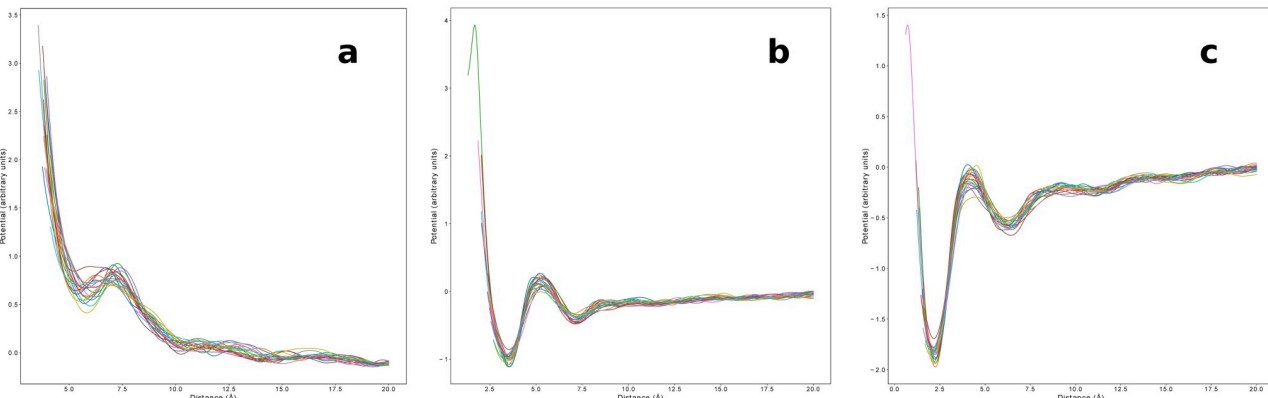

**Fig 4. Empirical knowledge-based potentials for different coarse-graining levels.** Effective potential $V_m(R)$ estimated for each $70 \leq m \leq 90$ in the Flory regime using Eq (9). Remarkably, the curves obtained with this procedure do not depend on the fragment length and can therefore be interpreted as an effective potential between the terminal fragment residues. In this case, where all fragments are considered regardless of the type of amino acids at their ends, the potential can be interpreted as a generic sequence independent interaction between all residues. (**a**) CA representation. (**b**) HV representation. (**c**) HH representation. Figure drawn with python package matplotlib, version 3.4.1. URL https://pypi.org/project/matplotlib/.

obtained with KDE using a kernel bandwidth $w$ (see Methods for details). Eq 9 highlights the dependence of such potential on the scale parameter $b$ used for the reference state and on the kernel bandwidth $w$ used to obtain the empirical distribution. It is worth noting that both parameters are obtained through a maximum likelihood approach.

We plot in Fig 4 the effective potentials $V_m(R)$ in the Flory regime $70 \leq m \leq 90$ in which $b(m) \simeq b^*$ is effectively uniform, for all coarse-graining levels used in this work. Remarkably, the curves for different fragment lengths $m$ collapse nicely together, allowing to recover well defined effective potentials $V^*(R)$ that we define as the average of the potentials obtained from Eq (9) over all fragment lengths $70 \leq m \leq 90$ in the Flory regime (the resulting average potentials are shown in S8 Fig in S1 File for all coarse-graining levels, along with the corresponding standard deviation). Such result pinpoints the existence of a robust underlying mechanism which is revealed by using the ratio between the empirical and the ideal reference distributions and which can allow for a consistent estimate of amino-acids interactions. Even more remarkably, we observe that while the empirical end-to-end distance distributions collapse upon rescaling (see Fig 3), deviations from the Maxwellian reference state collapse when rescaling back to the physical distance values (see Fig 4). For example, S7 Fig in S1 File clearly shows, for the HH and HV cases, how the position of sharp small peak at short distances, that determines the main minimum of the statistical potential $V^*(R)$, drifts upon changing $m$ when using rescaled distances.

The effective statistical potentials $V^*(R)$ differ significantly depending on the coarse-graining level, as in fact expected for physics-based interactions. The potentials obtained when considering all atoms, either with (HH) or without (HV) hydrogens share in fact similar features: a steep short range repulsive part and a series of well defined attractive minima with decreasing depth for increasing distance (see Table 2). At large distances the potential vanishes towards zero, although shallower minima can be still identified (see Table 2). However, in the more coarse-grained HV case, the first minimum is partially smoothed out and the barrier separating the first two minima becomes repulsive. In the even more coarse-grained CA case, the minima of the statistical potential get much more smoothed out and the potential becomes repulsive for all distances.

The positions and depths of the minima of the statistical potentials for different coarse-graining levels are reported in Table 2. Minima features are extracted using the $V^*(R)$ potential

**Table 2. Local minima features of the average effective statistical potential.**

| position (Å) | | | value ($\kappa_B T$) | | |
|---|---|---|---|---|---|
| **HH** | **HV** | **CA** | **HH** | **HV** | **CA** |
| 2.21 | 3.54 | 5.81 | −1.91 | −1.07 | 0.53 |
| 6.34 | 7.16 | 10.59 | −0.61 | −0.46 | 0.03 |
| 11.01 | 9.35 | 12.03 | −0.26 | −0.21 | 0.02 |
| 14.90 | 11.44 | 14.92 | −0, 13 | −0.21 | −0.08 |
| 15.69 | 15.90 | 19.55 | −0.13 | −0.12 | −0.13 |
| 18.95 | 17.63 | | −0.06 | −0.11 | |

Positions and values of the minima of the average effective statistical potential $V^*(R)$ for different coarse-graining levels.

obtained in the Flory regime $70 \leq m \leq 90$. We observe that the deepest minimum in the HH case (2.21 Å) corresponds to twice the Van der Waals radius 1.1 Å for the hydrogen atom [57], whereas the deepest minimum in the HV case (3.54 Å) is within the distance range observed between donor nitrogen and acceptor oxygen atoms for hydrogen bonds occurring in proteins [58].

In order to study the short distance repulsive behavior of the effective potential, more statistics is needed at very short distances. To this aim, we then consider the average $\overline{V}(R)$ of the statistical potentials defined by Eq (9), taken over the wider range of fragment lengths, $30 \leq m \leq 90$, for which the rescaled end-to-end distance distributions are close to Maxwellians (see Fig 2). The reference state is thus now defined with a non uniform Kuhn length $b(m)$. The quality of the collapse of the different $V_m(R)$, $30 \leq m \leq 90$, worsens, yet is still acceptable, as shown in S9 Fig in S1 File for all coarse-graining levels.

The sequence independent effective potential is plotted in logarithmic scale in Fig 5 for the CA case, together with a linear regression fit for the short range region.

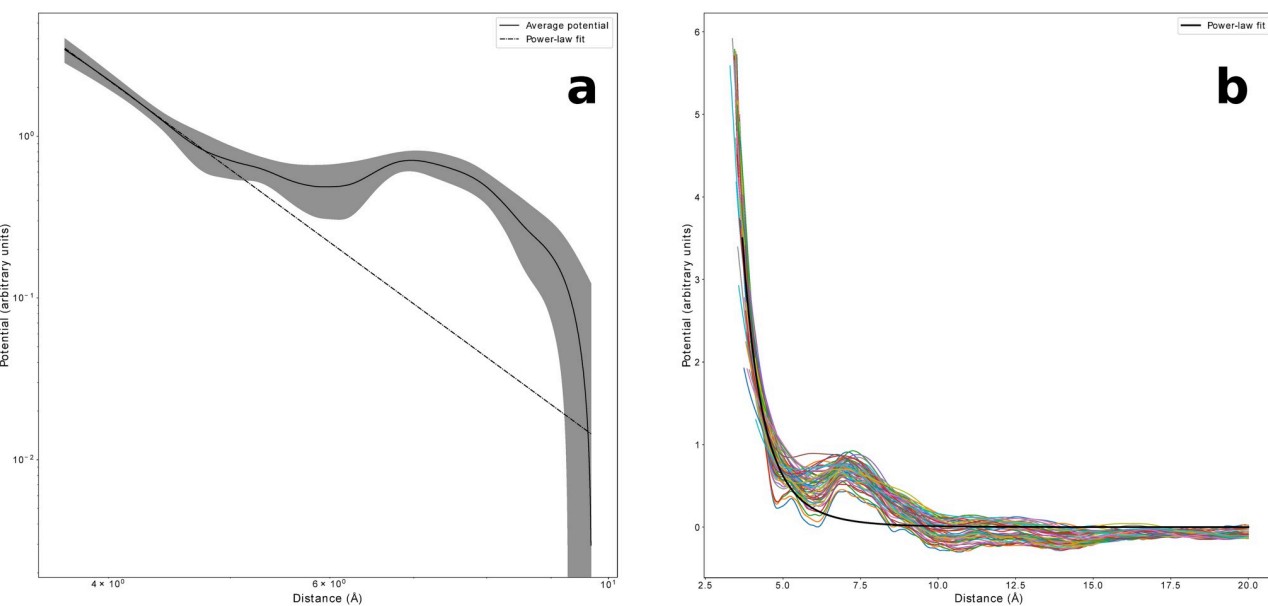

**Fig 5. Short distance behavior of the empirical knowledge-based potential.** Potential $\overline{V}(R)$ for the CA case, obtained when averaging the statistical potentials (9) over all fragment lengths $30 \leq m \leq 90$, shown together with the power-law fit at short distances. The exponent estimate is −5.7±0.3. (**a**) log-log scale; the standard deviation is also shown. (**b**) linear scale with data collapse of all $V_m(R)$ potentials for different values of sequence separation $30 \leq m \leq 90$. Figure drawn with python package matplotlib, version 3.4.1. URL https://pypi.org/project/matplotlib/.

The existence of a power law behavior seems clear. The resulting estimate for the exponent is −5.7±0.3, which might be related to the presence of distinctive dipole-dipole interactions. Nevertheless, we caution that the above exponent estimate may depend on the limited range of short distances that can be probed with the available statistics. As a matter of fact, the use of a dipole-based description for peptide groups was already successfully proposed to perform coarse grained simulations of protein folding [59].

### Statistical potentials with a Gaussian reference state: Sequence dependent effective interactions

The analysis carried out in the previous subsection can be repeated by splitting the full data set according to the specific amino acid types found at the end of the considered protein fragments. The resulting statistical potential should be interpreted as an effective interaction between the terminal residues. The decreased statistics, unfortunately, pushes our approach to its very limit, even when considering the average potential $\overline{V}(R)$ over all sequence separation values $30 \leq m \leq 90$ and a reference state with variable $b$.

For completeness, we report in Fig 6 some examples of average statistical potentials $\overline{V}(R)$ derived in our approach for the CA case, involving two cysteine residues (CYS-CYS), two small non-polar residues (ALA-ALA), two charged residues (GLU-GLU) and two hydrohobic (LEU-LEU) residues.

It clearly appears that the sequence-dependent potential can differ significantly from the average sequence-independent one. It is also interesting to notice how the obtained potentials reflect the physical-chemical properties of the amino acids. We indeed see that we obtain a strongly negative (attractive) interaction between two cysteines and a strongly repulsive one between two equally charged amino acids (GLU-GLU). The interactions between two small and non polar residues matches very closely the average behavior of the sequence-independent potential, whereas two hydrophobic residues, despite strongly repulsive at small distances, show an attractive interaction at longer distances. Similar plots for the HV and HH cases are shown in S10 and S11 Figs in S1 File, respectively.

### Discussion

As a first result of this paper, we have confirmed that the statistical properties of an ensemble of long enough fragments, collected from different globular proteins and selected to be buried in their interior, are similar to those of Gaussian ideal chains in a polymer melt [49]. The data set that we use [54] is based on experimentally derived protein native structures [27]. Fig 1a in fact shows that for sequence separations $70 \leq m \leq 90$ the average fragment end-to-end distance, computed between $C_\alpha$ atoms, scales as $m^{1/2}$, as expected for an ideal chain. At the same time, Fig 1b shows that the scale parameter $b(m)$, that maximizes the likelihood to the Maxwell distribution expected for ideal end-to-end distances, consistently plateaus to a uniform value $b^* = 3.67\pm0.01$ Å in the same sequence separation range.

On the other hand, Fig 2 shows that, even outside the $70 \leq m \leq 90$ range where the Flory theorem seems to hold, empirical distributions are reasonably approximated by Maxwell distributions in the whole sequence separation range $30 \leq m \leq 90$, with a scale parameter $b(m)$ non uniform for $m < 70$. In fact, both bounds for the larger range are only approximately determined in this work. The lower bound $m \gtrsim 30$ is necessary to avoid the local rigidity effects brought about mostly by secondary structure elements. The latter ones are otherwise seen to play a role for $m \lesssim 30$, resulting in a non zero tangent-tangent correlation (see S3 Fig in S1 File), not consistent with a Gaussian regime. The presence of secondary structure elements is instead fully compatible with the observation of ideal Gaussian statistics for longer

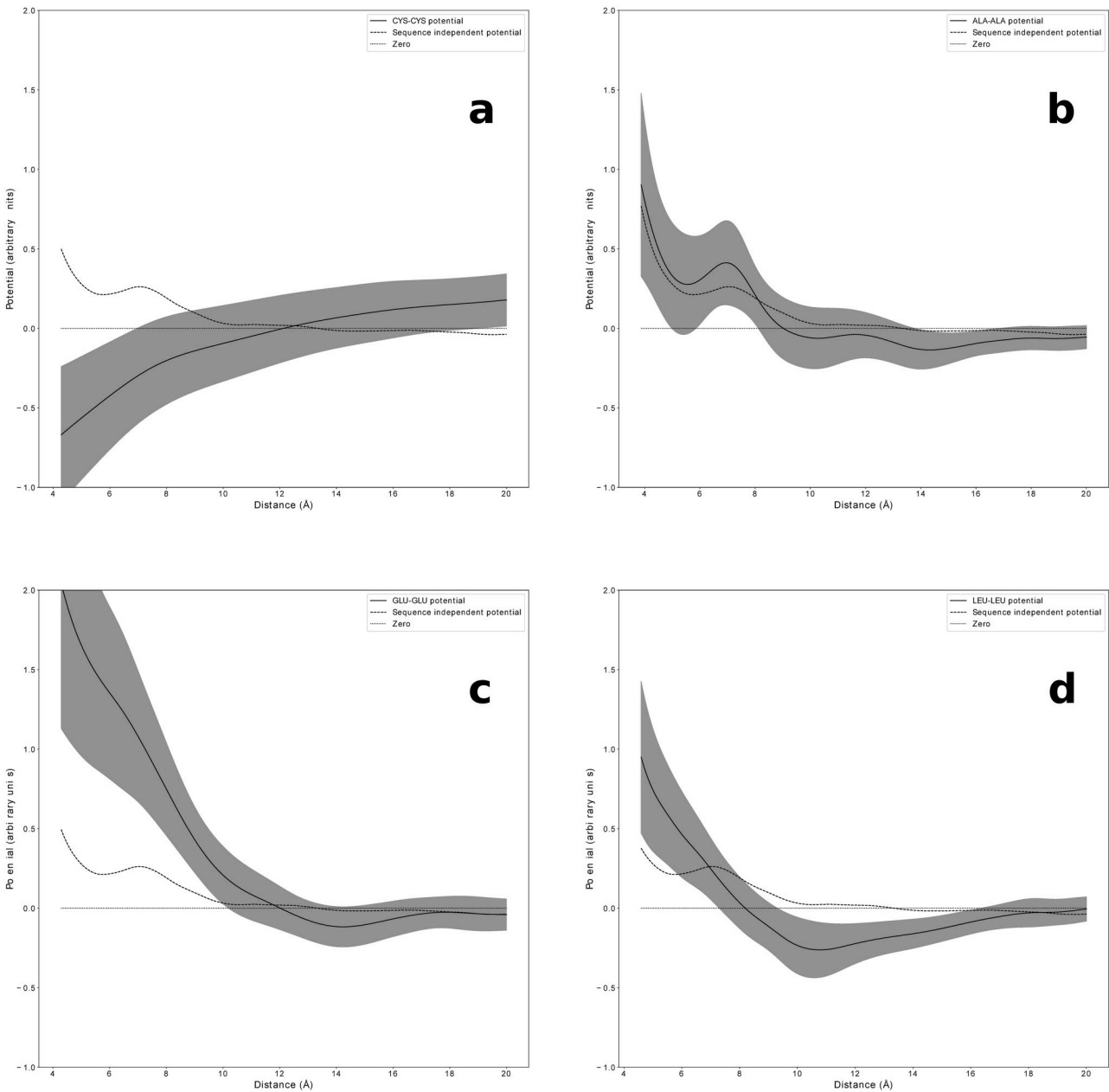

**Fig 6. Sequence-dependent empirical knowledge-based potentials.** Examples of sequence-dependent potentials $\overline{V}(R)$ for the CA case, obtained when averaging the statistical potentials (9) over all fragment lengths $30 \leq m \leq 90$, with its standard deviation (gray areas). The sequence independent potential (dashed line) is shown as a reference. (**a**) Cysteine-Cysteine (**b**) Small non polar residues ALA-ALA. (**c**) Two negatively charged residues GLU-GLU. (**d**) Two hydrohobic residues (LEU-LEU). Figure drawn with python package matplotlib, version 3.4.1. URL https://pypi.org/project/matplotlib/.

fragments. On the other hand, the upper bound $m \lesssim 90$ is due to the lack of statistics caused by the constraint $m < N^{2/3}$ for buried fragments combined with the available protein lengths in the dataset. Were longer proteins available, we would expect the Gaussian statistics to hold for even longer buried fragments. The intriguing observation that medium-sized buried protein fragments follow ideal chain statistics with varying Kuhn length is a novel result and we believe it deserves further investigation.

Fig 3 shows the remarkable data collapse of empirical end-to-end distance distributions in the Flory regime $70 \leq m \leq 90$, obtained upon rescaling $C_\alpha - C_\alpha$ distances by $m^{1/2}$. Notably, as shown in S4-S7 Figs in S1 File, we find similar results when computing fragment end-to-end distances with a more fine-grained representation of the protein chain, for either all atom (including hydrogen atoms, HH) or for all heavy atoms (excluding hydrogen atoms, HV). The observed ideal chain behaviour is due to the compensation between excluded volume effects and amino acids interactions, as predicted by the Flory theorem in a polymer melt. Our results extends previous findings based on a smaller data set of protein structures [49]. Moreover, we clearly show how the region in which the theorem applies should be determined.

The crucial novel observation that we make in our study is that, although the Gaussian statistics is valid for a large range of end-to-end distances, at short spatial scales there are deviations due to the fact that the excluded volume effect, as well as other interactions, cannot obviously fully disappear for real protein fragments. We exploit these deviations to extract effective interaction potentials between amino-acids at fragment ends by comparing the empirical probability distribution with the ideal one taken as a reference. The effective canceling out of different interactions, achieved in our reference state ensemble due to Gaussian statistics, allows us to estimate an unbiased physics-based pairwise interaction potential, without the spurious correlations present in general because of the chain constraint, of local conformational preferences, or of interior-exterior partitioning effects.

Following this approach, a different statistical potential can be estimated separately for any given value of sequence separation for which the ideal statistics is a good approximation of the empirical distribution. The main result of this work, shown in Fig 4, is the collapse of the different statistical potentials in the Flory regime $70 \leq m \leq 90$. Most remarkably, while the reference states for different sequence separations collapse when rescaling physical distances according to the ideal Gaussian scaling, see Fig 3, deviations from the reference states collapse when rescaling back to physical distances, see Fig 4, a strong hint that the statistical potentials that we estimate in this work do indeed capture physics-based effective interactions. As a consequence, short range deviations consistently behave as finite size corrections, drifting towards zero rescaled distance for longer and longer fragments, when the universal Gaussian behavior is eventually obtained in the limit of infinite fragment size. Even more remarkably, different potentials are obtained for the different coarse-graining levels used in this work, again as expected for physics-based effective interaction potentials.

For all coarse-graining levels, the statistical potential vanishes at large distances. Well defined local minima can be observed, listed in Table 2, with the deepest ones corresponding for the atomistic resolutions to steric (sum of Van der Waals radii) or hydrogen bonding interactions. The potential mimima get smoothed when considering a coarser representation, as expected for a proper coarse-graining when the finer degrees of freedom are averaged out. Within the $C_\alpha$ representation, the statistical potential is basically always repulsive; this can be rationalized by observing that the ideal Gaussian reference state already takes into account the average hydrophobic attraction needed for stabilizing a protein globule.

The short-range behaviour is not easy to investigate, since small values of end-to-end distance are scarcely sampled, and the use of shorter, more numerous, fragments is then required. Within the $C_\alpha$ representation, Fig 5 shows that a power law repulsion is found, with an exponent estimate consistent with −6. This could be related to the presence of peculiar dipole-dipole interactions.

Finally, we show in Fig 6 how the same approach can be used to derive sequence-dependent statistical potentials. Unfortunately the statistics available for buried protein fragments with a given pair of amino acid types at their end is barely enough to provide significant signals. Nonetheless, we observe trends consistent with what is expected from the physical-chemical

features of the probed types of residue pairs. The Gaussian reference state from buried protein fragments could be used, in principle, to estimate an orientation-dependent statistical potential, when needed to properly represent specific interaction modes, such as disulfide bonds between pairs of Cysteine residues. Available statistics would currently be a major issue, but this bottleneck is likely to be overcome in the near future due to the rapid increase of native structures being deposited in the PDB.

An ideal chain reference state was already used to define statistical potentials, in order to take into account chain connectivity in a minimal way [44]. However, our work shows that the use of an ideal chain reference state is well justified only for buried protein fragments, being in fact rooted into the non trivial polymer physics properties of protein globules. In fact, for fragment lengths $m \lesssim N$, well above the threshold $N^{2/3}$ used here to identify buried fragments, one expects to observe a non Gaussian behaviour characterized by the thermal exponent $\nu = 1/3$ typical of compact globules (see S2 Fig in S1 File for evidence of the "compact globule" scaling of gyration radius with protein length). We believe this finding may be an important conceptual advance.

For example, we can revisit one of the main criticisms raised against the standard derivation of statistical potentials [46, 47]. According to such critique, the Boltzmann inversion should be used in principle when averaging over different configurations from the thermal ensemble of the same protein system, not when averaging over a set of "fixed" configurations (native PDB structures) from different protein systems. However, it was observed that for some protein "substructures", the frequencies of different "states" in the PDB database correlate with what expected from thermodynamic behavior, although with different apparent temperatures. In particular, a main role is played by the apparent temperature associated to interior-exterior partitioning, which was shown to depend on the length, composition and compactness of the proteins in the database [46]. The finding that buried protein fragments collected from different protein systems do follow Gaussian statistics (the thermodynamic expected behavior for a polymer melt system) may at least be rationalized noting that by using only buried fragments the possible variation of the apparent temperature associated to interior-exterior partitioning does not play a role anymore.

We believe further work is needed to investigate in more detail the role of the constraint used to select buried protein fragments. In particular, relaxing that constraint, i.e. using $m < aN^{3/2}$ with $a \gtrsim 1$, could be a way to gather more statistics and obtain more reliable sequence-dependent potentials. A trade-off is at play, since the larger $a$, the more fragment statistics can be collected, but the less effective would be the constraint in selecting actually buried fragments. It is important to observe that the statistical potential presented here, in order to be tested in practical applications such as model quality assessment, should be necessarily complemented with other scoring terms, assessing for example solvent accessibilities and local conformational preferences. These properties are crucial for the correct folding of proteins and can not be detected in our reference state of buried fragments.

Moreover, we mention that it would be interesting to compare the results obtained here for buried fragments in protein globules, to the properties of fragments buried in the interior of polymer conformations sampled in the compact phase below the $\theta$-point. In particular, it is interesting to speculate whether the Gaussian behaviour with non uniform Kuhn length found here for intermediate size fragments is peculiar to proteins or not.

To conclude, we observe that the statistical properties uncovered in this work were derived analyzing ensembles built with different protein chains. Nonetheless, we may predict that the very same properties, reminiscent of ideal chain behaviour, could be observed for single protein chains in native conditions, for the specific case of Intrinsically Disordered Proteins (IDPs) that can form collapsed, globular ensembles while simultaneously exhibiting significant

conformational heterogeneity [60]. This prediction could be in principle tested by means of single molecule FRET experiments in which fluorescent labels can be placed across different chain fragments, thereby providing a direct measurement of end-to-end fragment distances [61]. Similar experiments were in fact already carried on for IDPs that form extended heterogeneous ensembles [62].

## Supporting information

**S1 File.**
(PDF)

## Author Contributions

**Conceptualization:** Stefano Zamuner, Flavio Seno, Antonio Trovato.

**Data curation:** Stefano Zamuner.

**Formal analysis:** Stefano Zamuner, Flavio Seno, Antonio Trovato.

**Investigation:** Stefano Zamuner.

**Methodology:** Stefano Zamuner, Flavio Seno, Antonio Trovato.

**Project administration:** Flavio Seno.

**Resources:** Antonio Trovato.

**Software:** Stefano Zamuner.

**Supervision:** Flavio Seno, Antonio Trovato.

**Validation:** Stefano Zamuner, Flavio Seno, Antonio Trovato.

**Visualization:** Stefano Zamuner, Flavio Seno, Antonio Trovato.

**Writing – original draft:** Stefano Zamuner, Flavio Seno, Antonio Trovato.

**Writing – review & editing:** Stefano Zamuner, Flavio Seno, Antonio Trovato.

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
