## [Decision Letter · Decision Letter 0]

16 Aug 2021

PONE-D-21-21648

Statistical potentials from the Gaussian scaling behaviour of chain fragments buried within protein globules

PLOS ONE

Dear Dr. Trovato,

Thank you for submitting your manuscript to PLOS ONE. After careful consideration, we feel that it has merit but does not fully meet PLOS ONE’s publication criteria as it currently stands. Therefore, we invite you to submit a revised version of the manuscript that addresses the points raised during the review process.

As you will read, the reviewers are mostly supportive of your manuscript, but still have a few requests for clarifications, and - to a lesser extent - requests for a quantified statistical perspective. I agree with them and I hope you can address these in your revised manuscript. 

We look forward to receiving your revised manuscript.

Kind regards,

Jerome Baudry, Ph.D.

Academic Editor

PLOS ONE

Journal Requirements:

Reviewers' comments:

Reviewer's Responses to Questions

**Comments to the Author**

1. Is the manuscript technically sound, and do the data support the conclusions?

Reviewer #1: Yes

Reviewer #2: Yes

2. Has the statistical analysis been performed appropriately and rigorously? 

Reviewer #1: Yes

Reviewer #2: Yes

3. Have the authors made all data underlying the findings in their manuscript fully available?

Reviewer #1: Yes

Reviewer #2: Yes

4. Is the manuscript presented in an intelligible fashion and written in standard English?

Reviewer #1: Yes

Reviewer #2: Yes

5. Review Comments to the Author

Reviewer #1: The study finds that long enough buried protein fragments follow Gaussian statistics, and further uses this Gaussian distribution as a reference state in the derivation of statistical potentials. This is an interesting and statistical mechanically rigorous way to define the reference state for statistical potentials. Overall, the manuscript is clearly written and the conclusion is supported by their data. I have a couple of general comments:

1. The Gaussian reference state is m-dependent. Would it be possible to define m-independent Gaussian reference states, at least for certain rang of m values?

2. For the m region whereby proteins behave like a random Gaussian chain, the use of a Gaussian reference state seems reasonable. What about the regions with small m values (m < 30) or large m values (m>90)?

Reviewer #2: The manuscript is well explained and interesting. I enjoyed reading it. A particular strength is that the limitations of the theoretical approach are described in detail. Some points for the authors to consider when revising the manuscript

- The abstract states that "Gaussian statistics [...] holds for a wide range of fragment lengths [...]". However, the authors note many times in the manuscript that they found deviations from Gaussian statistics at short spatial scales. The sentence in the Abstract should be modified to reflect more fully the data presented in the manuscript.

- Uncertainties ("error bars") are not provided for some of the fitted quantities. For example b*=3.67 A (line 242).

- How does the Kuhn length b*=3.67 A compare to the average distance between alpha carbons in proteins?

- It is stated i the Discussion that "[...] the use of an ideal chain reference state is well justified **only** for buried protein fragments," (I added * for emphasis). This statement is not backed by the data presented, the authors have examined only burried fragments and not non-burried/exposed fragments.

- Variable R is first introduced in line 130, but not defined there. R is defined later in line 148.

- Line 173: do the authors mean r.h.s. instead of l.h.s?

6. PLOS authors have the option to publish the peer review history of their article (what does this mean?). If published, this will include your full peer review and any attached files.

Reviewer #1: No

Reviewer #2: No

---

## [Author Response · Author response to Decision Letter 0]

30 Sep 2021

We thank both reviewers for their supportive and careful comments which helped us to improve our manuscript. Below a detailed point-by-point response to their comments.

Reviewer #1: The study finds that long enough buried protein fragments follow Gaussian statistics, and further uses this Gaussian distribution as a reference state in the derivation of statistical potentials. This is an interesting and statistical mechanically rigorous way to define the reference state for statistical potentials. Overall, the manuscript is clearly written and the conclusion is supported by their data.

We thank the reviewer for her/his positive evaluation.

I have a couple of general comments:

1. The Gaussian reference state is m-dependent. Would it be possible to define m-independent Gaussian reference states, at least for certain rang of m values?

We thank the reviewer for her/his observation. In fact, we realized that the description of how we define the “average” potential V(R) was possibly misleading and we corrected it in the revised version. As a matter of fact, we always compute V(R) as an average of potentials defined for different values of m, each having a different Gaussian reference state (i.e. a different Kuhn length b(m)). In the Flory regime (70 ≤ m ≤ 90) we could in fact use the same Gaussian reference state (with the “uniform” Kuhn length b* that we estimate in our manuscript) for all the potentials, as suggested by the referee. However, the resulting average potential would in practice be the same as the one computed with m-dependent reference states (see the following figures, where both average potentials are shown for all coarse-graining levels that we used in our work - figures visible in the pdf file). Finally, we realized that the Flory range was mistakenly labeled as 70 < m < 90 in the former version of the manuscript. We corrected this to 70 ≤ m ≤ 90 (and similarly to 30 ≤ m ≤ 90 for the overall range of fragment lengths) in the revised version.

2. For the m region whereby proteins behave like a random Gaussian chain, the use of a Gaussian reference state seems reasonable. What about the regions with small m values (m < 30) or large m values (m>90)?

We thank the reviewer for her/his observation. It is definitely worth discussing why we could not find a Gaussian reference state outside the 30 ≤ m ≤ 90 range. First, we acknowledge that both bounds for the larger range are only approximately determined in our work. The lower bound is necessary to avoid the local rigidity effects brought about mostly by secondary structure elements. The latter ones are otherwise seen to play a role for m < 30, resulting in a non zero tangent-tangent correlation (see S3 Fig), not consistent with a Gaussian regime. The presence of secondary structure elements is instead fully compatible with the observation of ideal Gaussian statistics for longer fragments. On the other hand, the upper bound m > 90 is due to the lack of statistics caused by the constraint m < N2/3 for buried fragments combined with the available protein lengths in the dataset. Were longer proteins available, we would expect the Gaussian statistics to hold for even longer buried fragments. We inserted a paragraph in the discussion section along the above lines.

Reviewer #2: The manuscript is well explained and interesting. I enjoyed reading it. A particular strength is that the limitations of the theoretical approach are described in detail. Some points for the authors to consider when revising the manuscript

We thank the reviewer for her/his positive evaluation and for her/his observations.

- The abstract states that "Gaussian statistics [...] holds for a wide range of fragment lengths [...]". However, the authors note many times in the manuscript that they found deviations from Gaussian statistics at short spatial scales. The sentence in the Abstract should be modified to reflect more fully the data presented in the manuscript.

We agree with the referee and we modified the Abstract accordingly.

- Uncertainties ("error bars") are not provided for some of the fitted quantities. For example b*=3.67 A (line 242).

We agree with the referee (and the editor) that error bars need to be provided. Accordingly, we updated Fig 1b and S4 Fig by showing the error bars associated to the maximum likelihood estimators b(m). Based on the Fisher information evaluated at b(m), standard deviations were estimated as (equattion visible in the pdf file) , where n(m) is the number of fragments in the dataset for a given m (see Table 1). In details, the variance of b(m) is estimated as the inverse of the Fisher information, the latter being minus the second derivative of the log-likelihood function with respect to the parameter b.

Moreover, we also provided the uncertainty for b* , estimated as the standard deviation of the 21 b(m) values (70 ≤ m ≤ 90) whose mean defines b* . For all coarse-graining levels the b* uncertainty turns out to be 0.01 Å. Such uncertainties were also represented in the updated Fig 1b and S4 Fig.

A proper explanation of how standard deviations for b(m) and b* was added in the revised manuscript in the Result section. 

- How does the Kuhn length b*=3.67 A compare to the average distance between alpha carbons in proteins?

We thank the reviewer for her/his observation. The Kuhn length is in fact close to the average distance, 3.8 Å, found between consecutive CA atoms in protein native structures. We added a sentence in the Result section.

- It is stated i the Discussion that "[...] the use of an ideal chain reference state is well justified **only** for buried protein fragments," (I added * for emphasis). This statement is not backed by the data presented, the authors have examined only burried fragments and not non-burried/exposed fragments.

We thank the reviewer for her/his observation. We agree that we did not examine the behavior of non-buried/exposed fragments. However, our statement can be backed by the following, admittedly indirect, argument. For fragment lengths m < N but of the same order as chain length N, that is well above the threshold N^(2/3) used here to identify buried fragments, one expects to observe a non Gaussian behaviour 

characterized by the thermal exponent ν = 1/3 typical of compact globules. In fact, we provide evidence of the “compact globule" scaling of gyration radius with protein length in S2 Fig. We inserted a sentence along the above lines in the Discussion section.

- Variable R is first introduced in line 130, but not defined there. R is

defined later in line 148.

The referee is correct, we introduced the quantity R in a confusing way. First, we now correctly states (lines 116-118 in Dataset subsection) that the gyration radius scaling is used to filter the protein dataset. Second we now introduce R as the end-to-end distance in line 130 (Materials and Methods section) and line 206 (Discussion section). 

- Line 173: do the authors mean r.h.s. instead of l.h.s?

Yes we mean r.h.s. Thanks for spotting this!

---

## [Decision Letter · Decision Letter 1]

29 Oct 2021

Statistical potentials from the Gaussian scaling behaviour of chain fragments buried within protein globules

PONE-D-21-21648R1

Dear Dr. Trovato,

We’re pleased to inform you that your manuscript has been judged scientifically suitable for publication and will be formally accepted for publication once it meets all outstanding technical requirements.

Kind regards,

Jerome Baudry, Ph.D.

Academic Editor

PLOS ONE

Additional Editor Comments (optional):

Reviewers' comments:

Reviewer's Responses to Questions

**Comments to the Author**

1. If the authors have adequately addressed your comments raised in a previous round of review and you feel that this manuscript is now acceptable for publication, you may indicate that here to bypass the “Comments to the Author” section, enter your conflict of interest statement in the “Confidential to Editor” section, and submit your "Accept" recommendation.

Reviewer #1: All comments have been addressed

2. Is the manuscript technically sound, and do the data support the conclusions?

Reviewer #1: Yes

3. Has the statistical analysis been performed appropriately and rigorously? 

Reviewer #1: Yes

4. Have the authors made all data underlying the findings in their manuscript fully available?

Reviewer #1: Yes

5. Is the manuscript presented in an intelligible fashion and written in standard English?

Reviewer #1: Yes

6. Review Comments to the Author

Reviewer #1: (No Response)

7. PLOS authors have the option to publish the peer review history of their article (what does this mean?). If published, this will include your full peer review and any attached files.

Reviewer #1: No

---

## [Editor Report · Acceptance letter]

7 Jan 2022

PONE-D-21-21648R1 

Statistical potentials from the Gaussian scaling behaviour of chain fragments buried within protein globules 

Dear Dr. Trovato:

I'm pleased to inform you that your manuscript has been deemed suitable for publication in PLOS ONE. Congratulations! Your manuscript is now with our production department. 

Kind regards, 

on behalf of

Dr. Jerome Baudry 

Academic Editor

PLOS ONE